# Oxaliplatin-Induced Damage to the Gastric Innervation: Role in Nausea and Vomiting

**DOI:** 10.3390/biom13020276

**Published:** 2023-02-01

**Authors:** Ahmed A. Rahman, Philenkosini Masango, Rhian Stavely, Paul Bertrand, Amanda Page, Kulmira Nurgali

**Affiliations:** 1Institute for Health & Sport, Victoria University, Melbourne, VIC 3021, Australia; 2Department of Pediatric Surgery, Massachusetts General Hospital, Harvard Medical School, Boston, MA 02114, USA; 3College of Health & Biomedicine, Victoria University, Melbourne, VIC 3011, Australia; 4School of Health and Biomedical Sciences, Royal Melbourne Institute of Technology University, Melbourne, VIC 3083, Australia; 5Vagal Afferent Research Group, Centre for Nutrition and Gastrointestinal Disease, Adelaide Medical School, University of Adelaide, Adelaide, SA 5005, Australia; 6Department of Medicine Western Health, The University of Melbourne, Melbourne, VIC 3010, Australia; 7Regenerative Medicine and Stem Cells Program, Australian Institute for Musculoskeletal Science (AIMSS), Melbourne, VIC 3021, Australia

**Keywords:** chemotherapy, oxaliplatin, stomach innervation, enteric neurons, emetic reflex, vagus nerve

## Abstract

Nausea and vomiting are common gastrointestinal side effects of oxaliplatin chemotherapy used for the treatment of colorectal cancer. However, the mechanism underlying oxaliplatin-induced nausea and vomiting is unknown. The stomach is involved in the emetic reflex but no study investigated the effects of oxaliplatin treatment on the stomach. In this study, the in vivo effects of oxaliplatin treatment on eating behaviour, stomach content, intrinsic gastric neuronal population, extrinsic innervation to the stomach, levels of mucosal serotonin (5-hydroxytryptamine, 5-HT), and parasympathetic vagal efferent nerve activity were analysed. Chronic systemic oxaliplatin treatment in mice resulted in pica, indicated by increased kaolin consumption and a reduction in body weight. Oxaliplatin treatment significantly increased the stomach weight and content. The total number of myenteric and nitric oxide synthase-immunoreactive neurons as well as the density of sympathetic, parasympathetic, and sensory fibres in the stomach were decreased significantly with oxaliplatin treatment. Oxaliplatin treatment significantly increased the levels in mucosal 5-HT and the number of enterochromaffin-like cells. Chronic oxaliplatin treatment also caused a significant increase in the vagal efferent nerve activity. The findings of this study indicate that oxaliplatin exposure has adverse effects on multiple components of gastric innervation, which could be responsible for pica and gastric dysmotility.

## 1. Introduction

Colorectal cancer (CRC) is the third most common type of cancer, making up about 10% of all cases [1]. Risk factors include an older age, diet, obesity, smoking, and a lack of physical exercise [2,3]. Surgery, alone or in combination with chemotherapy, is the main and most common treatment strategy for CRC. Platinum-based chemotherapeutic drugs, including cisplatin, carboplatin, and oxaliplatin, are most commonly and preferably used for the treatment of CRC compared to non-platinum regimens due to the increased patients’ survival rate [4]. Oxaliplatin, the third-generation platinum compound, is now considered the safest chemotherapeutic agent for CRC treatment as it shows fewer adverse effects compared to cisplatin and carboplatin. Unfortunately, the benefit of this frequently prescribed drug is compromised by some severe side effects, including peripheral neurotoxicity, nausea, vomiting, constipation, and/or diarrhea [5].

Nausea and vomiting are two of the most troubling side effects experienced by patients during CRC chemotherapy. Approximately 70–80% of patients experience nausea and vomiting during and after chemotherapy [6]. Chemotherapy-induced nausea and vomiting have been classified into three categories: acute (occurs within 24 h of chemotherapy), delayed (occurs between 24 h and 5 days after treatment), and anticipatory (triggered by taste, odor, memories, visions, or anxiety related to chemotherapy) [7]. Conventional and available antiemetic agents used alone or in combination, include corticosteroids, 5-hydroxytryptamine (5-HT)_3_ receptor antagonists, and neurokinin (NK)-1 receptor antagonists [8,9]. Despite significant advancement in the treatment of chemotherapy-induced nausea and vomiting, these side effects remain a substantial problem for CRC patients and often lead to a delay or cessation of chemotherapy treatments [9]. In addition to poor chemotherapy adherence, these side effects contribute to increased anxiety and depression, and ultimately to a diminished quality of life of the patients. Therefore, research to enhance our understanding of the mechanisms of chemotherapy-induced nausea and vomiting is essential to develop novel drugs to improve treatment adherence and positive clinical outcomes for CRC patients.

Vomiting (or emesis) is a natural defense mechanism that includes afferent input into the vomiting centre located in the medulla, the integration of the signal in the vomiting centre, and the subsequent relay of necessary efferent output, leading to coordinated respiratory, gastrointestinal (GI), and abdominal function resulting in vomiting. The vomiting centre is predominantly activated by three different mechanisms: (i) excitatory nervous impulses from the stomach or intestinal tract resulting in a reflexive activation; (ii) signalling from the higher brain centres; or (iii) stimulation of the chemoreceptor trigger zone (CTZ) by noxious stimuli [8]. The stomach is innervated by vagal afferent fibres that contain 5-HT_3_ and NK-1 receptors [10], the two most important receptors involved in the induction of emesis. Activation of the enterochromaffin-like (ECL) cells of the gastric mucosa releases serotonin (5-HT) and NK-1 that can trigger the afferent vagus nerve to stimulate the CTZ and initiate emesis [11]. Parasympathetic vagal efferent nerves, innervating the GI tract, are activated during the emetic reflex, and relay the integrated neuronal response back to the upper GI tract, including the stomach, that ultimately results in the expulsion of gastric contents (emesis) (for a detailed review see [10,12]). Currently, the exact mechanisms of nausea and vomiting induced by oxaliplatin are unresolved.

Potential trigger points for oxaliplatin-induced nausea and vomiting could include alterations in the neurotransmitter release or damage to the intrinsic enteric and extrinsic autonomic innervation of the stomach. Chemotherapy causes irritation of the GI mucosa and it is now widely known that cytotoxic chemotherapeutic agents, particularly cisplatin, increase the release of 5-HT from the enteroendocrine cells and activate 5-HT_3_ receptors on the vagal afferents [10]. However, no study investigated the role of oxaliplatin in any part of the emetic reflex pathway, including gastric innervation, the release of 5-HT inducing emesis, and vagal efferent nerve activity relaying the signal for emesis. We hypothesize that oxaliplatin alters the neurotransmitter release, including 5-HT in the stomach, by damaging the mucosa as well as its intrinsic neuronal population, and parasympathetic nerve activity, leading to emesis. The aim of this study was to reveal oxaliplatin-induced changes in the gastric innervation providing a better understanding of the underlying mechanisms of chemotherapy-induced nausea and vomiting.

## 2. Materials and Methods

### 2.1. The Animals

All animal experiments of the study were performed in accordance with guidelines of the Australian Code of Practice for the Care and Use of Animals for Scientific Purposes and were approved by the Victoria University Animal Experimentation Ethics Committee. In this study male Balb/c mice aged 5–8 weeks (18–25 g, n = 30) were used for pica, morphological, and electrochemistry experiments. Male Sprague Dawley (SD) rats (age: 10–12 weeks, 350–450 g, n = 10) were used for electrophysiology experiments. All mice and rats had free access to food and water and were housed in a temperature-controlled environment with 12-h day/night cycles in the animal holding room at the Western Centre for Health, Research and Education (Melbourne, Australia). They were allowed to acclimatize for at least 7 days before experimental manipulations.

### 2.2. In Vivo Oxaliplatin Treatment

Mice and rats used in this study were randomly divided into two groups: oxaliplatin-treated and sham-treated. In the oxaliplatin-treated group, animals received intraperitoneal (i.p.) injections of oxaliplatin (3 mg/kg, per dose, dissolved in sterile water), 3 times a week via a 26-gauge needle. The dose of oxaliplatin (Tocris Bioscience, Bristol, UK) was calculated per body surface area to be equivalent to clinically used standard human doses [13]. In the sham-treated group, animals received sterile water via an i.p. injection 3 times a week via a 26-gauge needle (the maximum volume was 200 µL). Mice were weighed daily and monitored for signs of pica (ingestion of non-nutritive substances, for example, bedding materials) during the treatment regimen. Mice were euthanized by cervical dislocation 3, 7, or 14 days after the start of treatment. Stomach specimens were collected for immunohistochemical analyses. Multiple time points were analysed to study the progress of neuronal damage in response to treatment over time.

### 2.3. Assessment of Feeding Behaviour: Pica

The behavioural phenomena of oxaliplatin-induced nausea and vomiting were evaluated via measurement of the pica response. Mice do not vomit but show an equivalent measurable behaviour called pica, defined as eating non-food items such as bedding material. Thus, pica is an indirect marker of emesis that was measured by the ingestion of kaolin clay (a non-nutritive substance). Mice ingest kaolin as an innate curative response for GI disturbances. Oxaliplatin-induced pica was quantified by measuring the consumption of kaolin pellets. Kaolin pellets were prepared as previously described [14,15]. Briefly, kaolin (hydrated aluminum silicate; 98.5%) was mixed with 0.5% carmine and 1% gum Arabic to form a thick paste. The paste was then processed such that the shape and size of the kaolin pellets resembled that of the normal laboratory mouse food pellets.

After three days of acclimatization to the presence of kaolin (day 0), mice received i.p. injections either of oxaliplatin or saline, with kaolin and normal feed given immediately after i.p. injections. The amount of kaolin versus food intake was measured daily to observe the presence of abnormal eating patterns. Feces were collected at 0, 3, 7, and 14 days. After collection, fecal samples were counted and homogenised to determine the carmine concentration at absorbances of 550 and 700 nm using a spectrophotometer (PharmaSpec UV1700, Shimadzu, Japan).

### 2.4. Stomach Contents

Stomach weight was measured immediately upon collection from sacrificed mice. Following the measurement of the stomach weight, the stomach was opened via cutting along the greater and lesser curvature borders and the stomach contents were collected to weigh the stomach tissue and the dry weight of the stomach’s solid contents independently.

### 2.5. Tissue Preparation

After weighing and dissection, tissues were immediately placed in oxygenated phosphate-buffered saline (PBS; 0.1 M, pH = 7.2) containing an L-type Ca^2+^ channel blocker, nicardipine (3 μM; Sigma-Aldrich, Sydney, Australia) to inhibit smooth muscle contraction. Tissues were then processed in three different ways: (1) haematoxylin and eosin (H&E) staining for histology, (2) wholemount longitudinal muscle-myenteric plexus (LMMP) preparations, and (3) cross-sections. All tissues were pinned flat onto Sylgard-lined Petri dishes and fixed overnight at 4 °C in appropriate fixatives.

Histology with haematoxylin and eosin (H&E) staining. The stomach, including the fundus, corpus, and pylorus, was used for histology. After fixation in 10% buffered formalin, regions of the stomach were paraffin-embedded. This was followed by microtome sectioning at 5 μm of thickness. Tissue sections were deparaffinized, cleared, and then rehydrated in graded concentrations of ethanol. Tissue sections were immersed sequentially for particular time periods in xylene (4 min × 3 times), 100% ethanol (3 min), 90% ethanol (2 min), and 70% ethanol (2 min). Tissue sections were rinsed in tap water, and then immersed in hematoxylin for 4 min, again rinsed in tap water, then immersed in Scott’s tap water for 1 min and eosin for 6 min. After rinsing in tap water, tissue sections were immersed in 100% ethanol (1 min × 2 times) and xylene (3 min × 2 times). Sections were mounted on glass slides using a DPX mounting medium.

Wholemount LMMP preparations. The gastric corpus was dissected to expose the myenteric plexus by removing the mucosa and circular muscle layers under a stereomicroscope (Nikon SMZ 1000, Tokyo, Japan). After the dissection and fixation in Zamboni’s fixative (2% formaldehyde and 0.2% picric acid), tissues were washed in dimethyl sulfoxide (DMSO; Sigma-Aldrich) (10 min × 3 times) and in 0.1 M PBS (10 min × 3 times) to remove the fixative [16,17].

Cross-sections. Tissues for cross-sections were pinned in a Sylgard-lined Petri dish without stretching. After fixation in Zamboni’s fixative overnight at 4 °C, tissues were kept in 30% sucrose/PBS overnight at 4 °C for cryo-protection, and then placed into a 50% Optimal Cutting Temperature compound (OCT) in 30% sucrose/PBS for 12 h, followed by block-freezing in 100% OCT. Frozen tissues were sectioned at 20 μm thickness across the stomach width using a cryostat and mounted onto glass slides.

### 2.6. Immunohistochemistry

Immunohistochemistry (IHC) was performed on wholemount LMMP preparations and cross-sections of the gastric corpus. After pre-incubation in 10% normal donkey serum (NDS) (MilliporeSigma, Burlington, MA, USA) for 1 h at room temperature, the primary antibodies against the protein gene product (PGP) 9.5 (chicken; 1:500; Abcam, Cambridge, UK), vesicular acetylcholine transporter (VAChT; goat; 1:500 dilution; Merck Millipore, Macquarie Park, NSW 2113, Australia), neuronal nitric oxide synthase (nNOS; goat; 1:500 dilution; Sapphire Bioscience, Redfern, NSW 2016, Australia), calcitonin gene-related peptide (CGRP; rabbit; 1:3000 dilution; Sigma-Aldrich, St. Louis, MO 68178, USA), and tyrosine hydroxylase (TH; sheep; 1:1000 dilution; Merck Millipore, Macquarie Park, NSW 2113, Australia) were used for overnight incubation at 4 °C. The anti-PGP9.5 antibody is used as a marker for labeling all neuronal cell bodies [18]; the anti-nNOS antibody identifies predominantly inhibitory muscle motor neurons and some interneurons [19]; the anti-CGRP antibody is used as a marker for sensory afferent fibers [20], and intrinsic primary afferents [21]; the anti-VAChT antibody identifies cholinergic fibers [19,22]; and antibody to TH (rate-limiting enzyme in catecholamine synthesis) identifies noradrenergic fibers [23].

Following the incubation with primary antibodies, tissues were washed with 0.1 M PBS-Triton (10 min × 3 times) and were incubated in the corresponding secondary antibodies for 2 h at room temperature. The secondary antibodies Alexa Fluor 594 (donkey anti-chicken; 1:200 dilution; Jackson ImmunoResearch, West Grove, PA 19390, USA), Alexa Fluor 488 (donkey anti-goat; 1:200 dilution; Jackson ImmunoResearch), and fluorescein isothiocyanate (FITC; donkey anti-sheep; 1:200 dilution; Abacus, Mitcham, VIC 3132, Australia) were used. Then tissues were washed with PBS-Triton (10 min × 3 times) and mounted onto glass slides with a fluorescent mounting medium and coverslipped.

For ECL cell counting, tissues were immersed in a fixative of 4% paraformaldehyde and 3% sucrose in 0.1 M PBS for 4 h at 4 °C. The tissues were permeabilised in DMSO (10 min × 3 times), washed with 0.1 M PBS (10 min × 3 times), and embedded in the OCT compound. The tissue sections (12 μm thickness) were allowed to thaw and were then incubated with 10% NDS and 0.5% Triton X-100 diluted in 0.1 M PBS at room temperature to prevent nonspecific antibody binding in subsequent immunolabelling. After washing with 0.1 M PBS (10 min × 3 times), tissue sections were incubated with a rabbit anti-5-HT primary antibody (1:5000 dilution; Immunostar, Hudson, WI 54016, USA) and 2% NDS overnight at 4 °C. Tissue sections were washed as described above and incubated with a secondary antibody donkey anti-rabbit IgG Alexa Fluor 647 and 2% NDS for 1 h at room temperature. Tissue sections were counterstained with a nuclear marker, DAPI, for 2 min and washed prior to being mounted and coverslipped for imaging.

### 2.7. Imaging

Histological sections were visualised using a BX53 Olympus microscope (Evident Corporation, Tokyo, Japan), and images were captured with CellSens V3.2 software. Immunohistochemical images were captured with a Nikon Eclipse Ti multichannel confocal laser scanning system (Nikon, Tokyo, Japan). Double-labelled tissues were visualised and imaged by using combinations of filters appropriate for specific fluorophores. Images (512 × 512 pixels) were obtained using ×20 (dry, 0.75) or ×40 (oil immersion, 1.3) magnification. Neuronal structures were imaged by collecting Z-series of ten consecutive optical sections at 1 μm intervals. The 5-HT immunoreactivity was pseudo-coloured red for greater visual distinction against DAPI. All images were analysed with Image J software (National Institute of Health, Bethesda, MD 20892, USA).

### 2.8. Quantitative Analysis of Immunoreactivity and Histology

The immunoreactivity of PGP9.5 and nNOS-IR neurons in the myenteric plexus was assessed by analysing the number of immunoreactive neurons within a 0.25 mm^2^ area by randomly capturing 8 images per preparation at ×20 magnifications and averaged (the total area was 2 mm^2^). The density of the nerve fibre immunoreactivity in cross-sections of the stomach was also measured as above. All images were captured under identical acquisition exposure time conditions, calibrated to standardised minimum baseline fluorescence, and converted to binary. All images were analysed blindly. The area of the immunoreactive (IR) fibres was then expressed as a percentage of the total examined area. The number of ECL cells and the number of gastric pits were quantified in 8–10 non-adjacent fields per individual sample. From this the average number of ECL cells per gastric pit was calculated.

### 2.9. The 5-HT Measurements

Gastric 5-HT was measured using by amperometric recordings as previously described [24]. Gastric tissues were cut along the mesenteric border and pinned in a silicon-lined recording chamber containing carbogenated physiological Kreb’s solution at 35 °C, which was replaced at a flow rate of ~5 mL/min. Tissues were allowed to equilibrate for 60 min before 5-HT oxidation measurements were performed using carbon fibre microelectrodes voltage clamped at +400 mV. Recordings of the current generated by the oxidation of the 5-HT were presented as a positive deflection. Recordings were performed using a VA-10 amplifier (NPI electronic, 71732 Tamm, Germany), digitised at 1–5 kHz to a computer using the PC lamp 9.0 software with the settings: 0.5 kHz filtering with a 50 Hz notch filter. Electrodes were individually calibrated with a 10 μL spritz of 10 μM serotonin hydrochloride before recording from the gastric tissues. A precision micromanipulator (World Precision Instruments, Sarasota, FL, USA) was used to compress the mucosa with the carbon fibre microelectrode to record a mechanically stimulated 5-HT release (‘peak’) and the decay of 5-HT oxidation back to baseline levels (‘steady state’). The micromanipulator provides fine mechanical placement of the microelectrode with movement as little as a micron at a time. Therefore, the force applied to compress the mucosa was consistent. Moreover, the same investigator conducted all experiments in a blinded manner.

### 2.10. Electrophysiological Experiments

Electrophysiological experiments were conducted as described previously [25]. Briefly, rats were randomly divided into two groups (n = 10/group) and treated in the same manner as described in the mice experiments. Rats were anesthetized with urethane (1.2–1.4 g/kg, i.p.) on day 14 after the start of treatment. The level of anesthesia and body temperature were maintained with a supplemental dose of urethane and a heating pad, respectively. The left jugular vein was cannulated for the administration of drugs and fluids and the right carotid artery was cannulated for the measurement of blood pressure (BP). The trachea was cannulated to enable artificial ventilation and a 3-lead electrocardiogram was fitted. The vagus nerve was isolated, tied with a silk thread, and cut distally to permit the recording of efferent vagus nerve activity (VNA). Rats were secured in a stereotaxic frame, paralysed (pancuronium bromide; 0.8 mg initially, then 0.4 mg/h) and artificially ventilated with oxygen-enriched room air. Animals were kept hydrated by infusing 5% glucose intravenously. Bipolar silver wire electrodes were used for nerve recordings. The neurograms were amplified (×10,000), bandpass filtered (0.1–2 kHz), and sampled at 3 kHz. Recordings were made using the Spike2 software (v7.1, CED Ltd., Milton, Cambridge, UK).

### 2.11. Statistical Analysis

The analysis was conducted with a GraphPad Prism (version 7.0, GraphPad, La Jolla, CA 92037, USA). All values are expressed as means ± standard error. The unpaired *t*-test was used to analyse the difference in the number of ECL cells, the number of neurons per area, the percentage of the area of fibre expression between all groups, and the parameters of electrophysiology experiments. One-way analysis of variance followed by Bonferroni’s multiple group comparison test was used to analyse the weight of the stomach and its contents at different time points. The value *p* < 0.05 was considered to be significant.

## 3. Results

### 3.1. Effects of Oxaliplatin Treatment on Pica, Body Weight, and Stomach Content

Symptoms of GI side effects in mice including pica behaviour and body weight were recorded over the treatment period of 14 days after the first injection of oxaliplatin. The amount of food (normal chow) as well as kaolin consumption, the weight of the full stomach, empty stomach, and stomach content were measured at day 0 (prior to injection of oxaliplatin or sterile water), and at day 3, 7, and 14 post-treatment (after injection of oxaliplatin or sterile water). Weights of the full stomach, empty stomach, and stomach content were measured on days 3, 7, and 14 post-treatment.

Treatment with oxaliplatin (3 mg/kg, i.p.) caused a significant reduction in food intake at day 3 (*p* < 0.05), 7 (*p* < 0.001), and 14 (*p* < 0.001) post-treatment compared to sham-treated mice (n = 10 mice/group, Figure 1A). All mice had the same body weight and age at day 0. Conversely, the consumption of kaolin was significantly increased (*p* < 0.001) at days 3, 7, and 14 post-treatment in oxaliplatin-treated mice as compared to sham-treated mice (n = 10 mice/group, Figure 1B).

Mice were weighed daily prior to and post-oxaliplatin injection over a 14-day period. There was no significant difference between the sham and oxaliplatin-treated groups at day 0. Oxaliplatin administration caused a significant decrease in the body weight gain starting from day 3 after the first injection and throughout the following experimental period compared to sham-treated mice (n = 10 mice/group, *p* < 0.05, *p* < 0.01, and *p* < 0.001, Figure 1C). The deviation in the body weight within each group at various time points was not significant.

The effects of oxaliplatin on the stomach content were observed. Following i.p. injections over 14 days, the weight of the full stomach in the oxaliplatin-treated group was significantly increased (*p* < 0.001, n = 10 mice/group) at 7 and 14 days as compared to the sham-treated group (Figure 1D). Conversely, the weight of the empty stomach was significantly decreased (*p* < 0.001, n = 10 mice/group) in oxaliplatin-treated mice at days 7 and 14 as compared to sham-treated mice (Figure 1D). Stomach content weight was significantly increased (*p* < 0.001, n = 10 mice/group) following oxaliplatin treatment at days 7 and 14 when compared with sham-treated mice (Figure 1D).

### 3.2. Effects of Oxaliplatin Treatment in the Morphology of the Stomach

The effects of oxaliplatin on the morphological structure of the stomach were observed over a 14-day period. Muscle hypertrophy, disorganisation and distortion of gastric glands and pits, and epithelial exfoliation were observed in the corpus and pylorus of oxaliplatin-treated mice (n = 4/group, Figure 2). In the fundus, a loss of connective tissue and slight hypertrophy of the muscle layer were observed in the oxaliplatin-treated mice as compared to sham-treated mice (Figure 2A). Oxaliplatin-treated mice had a loss or distortion of the gastric surface epithelium and gastric pits of the corpus. Repeated administration of oxaliplatin caused the disorganisation of gastric glands with muscle hypertrophy (Figure 2B). In the pylorus, epithelial cell disorganisation and goblet cell distortion with severe muscle hypertrophy was observed in oxaliplatin-treated mice (Figure 2C).

### 3.3. Effects of Oxaliplatin Treatment on the Myenteric Neurons and Subpopulation of nNOS-IR Neurons in the Stomach

The total number of neurons in wholemount LMMP preparations of the gastric corpus was counted using the pan-neuronal marker PGP9.5. Repeated oxaliplatin administration caused a substantial decrease in the number of neurons per area (0.25 mm^2^) within the myenteric plexus of the stomach at day 7 (81 ± 4) and 14 (62 ± 6) (n = 4 mice/group) compared to the sham group (day 7: 108 ± 3; day 14: 103 ± 5; *p* < 0.001; n = 4 mice/group, Figure 3A,B).

The number of nNOS-IR neurons in wholemount LMMP preparations of the gastric corpus was quantified following immunohistochemical staining with the anti-nNOS antibody. Oxaliplatin treatment caused a significant decrease in the number of nNOS-IR neurons per unit of surface area (0.25 mm^2^) within the myenteric plexus of the stomach at day 3 (55 ± 3), 7 (41 ± 1), and 14 (34 ± 2) as compared to sham-treated mice (day 3: 68 ± 2; day 7: 64 ± 2; day 14: 62 ± 2; *p* < 0.05; *p* < 0.001; n = 4 mice/group; Figure 3C,D).

### 3.4. Effects of Oxaliplatin Treatment on the Expression of Cholinergic Fibres in the Stomach and Vagal Efferent Nerve Activity

The density of VAChT-IR cholinergic fibres was analysed using the anti-VAChT antibody in both cross-sections and wholemount LMMP preparations of the gastric corpus. In wholemount LMMP preparations, oxaliplatin treatment resulted in a significant reduction in the density of cholinergic fibres at day 7 (11 ± 0.8%) and 14 (6 ± 0.6%) compared to saline treatment (day 7: 14 ± 0.8%; day 14: 17 ± 1.0%; *p* < 0.05; *p* < 0.001; n = 4 mice/group; Figure 4A,B).

In the cross-section preparation, the density of VAChT-IR fibres projecting to the stomach mucosa displayed a significant decrease in the stomach of oxaliplatin-treated mice (5.5 ± 0.2%, n = 4 mice/group) as compared to sham-treated mice (7.2 ± 0.7%, *p* < 0.05, n = 4 mice/group, Figure 4C,D) at day 14.

In the functional electrophysiology experiments, rat vagal efferent nerve activity was recorded and analysed to evaluate the effect of oxaliplatin treatment on parasympathetic nerve activity. Chronic oxaliplatin treatment (n = 5 rats/group) significantly increased baseline VNf (64 ± 3.3 bpm vs. 49 ± 4.3 bpm, *p* < 0.05), but reduced baseline VNamp (2.4 ± 0.4 a.u. vs. 3.7 ± 0.6 a.u., *p* < 0.05) as compared to the sham treatment (n = 5 rats/group) (Figure 5).

### 3.5. Effects of Oxaliplatin Treatment on Noradrenergic and Sensory Nerve Fibres in the Stomach

The density of TH-IR fibres was evaluated in both cross-sections and wholemount LMMP preparations of the gastric corpus to determine changes in the expression of sympathetic noradrenergic fibres (Figure 6A–C). A significant decrease in the density of the TH-IR fibres in the myenteric plexus was observed at day 3 (5.0 ± 0.7%, *p* < 0.001), day 7 (5.0 ± 1.0%, *p* < 0.01), and day 14 (4.0 ± 0.2%, *p* < 0.001) in wholemount LMMP preparations of the stomach extracted from the oxaliplatin-treated group compared to the sham-treated group (day 3: 8.5 ± 0.7%; day 7: 7.5 ± 0.7%; day 14: 9.0 ± 0.8%; n = 4 mice/group; Figure 6A,C). In cross-section preparations, oxaliplatin treatment displayed a significant decrease in the percentage of area of the TH-IR fibres (2.8 ± 0.9%) projecting to the stomach mucosa compared to the sham-treated mice (4.0 ± 0.9%; *p* < 0.05; n = 4 mice/group, Figure 6B,D) by day 14.

Immunolabelling using the anti-CGRP antibody was performed to assess sensory (CGRP-IR) nerve fibres in the gastric corpus. CGRP-IR nerve fibres were distributed throughout the mucosa and myenteric plexuses of the gastric corpus. Repeated oxaliplatin administration caused a significant decrease in the percentage area of CGRP-IR fibres at day 7 (2.4 ± 0.3%, *p* < 0.01) and 14 (1.8 ± 0.6%, *p* < 0.05) in wholemount LMMP preparations of the stomach compared to the sham-treated group (day 7: 4.3 ± 0.2%; day 14: 4.3 ± 0.05%; n = 4 mice/group; Figure 6E,F). However, in cross-section preparations of the gastric corpus, there was no significant difference in the percentage of area of the CGRP-IR fibres projecting to the stomach between oxaliplatin-treated mice and sham-treated mice (data not shown).

### 3.6. Effects of Oxaliplatin Treatment on Serotonin Level in the Stomach

Extracellular 5-HT oxidation currents were measured spatially and temporally in the gastric mucosa in oxaliplatin-treated and sham-treated mice by electrochemistry techniques (Figure 7). ECL cells in the mucosal crypts were compressed by carbon fibre recording electrodes to induce a momentary peak in the 5-HT release (peak) before returning to basal levels (steady state) (Figure 7A). The quantification of 5-HT in the gastric mucosa revealed a significant elevation of the peak and steady state levels in oxaliplatin-treated mice (peak, 16.3 ± 1.6 µM; steady state, 7.1 ± 1.1 µM) compared to sham-treated mice (peak, 6.3 ± 1.2 µM; steady state, 3.5 ± 0.6 µM) (*p* < 0.01 for both) (n = 4 mice/group, Figure 7B).

To determine the cause of the increased 5-HT concentrations with oxaliplatin treatment, ECL cells were quantified by immunohistochemical labelling of vesicular 5-HT stored in mucosal cell bodies (Figure 7C). To identify their potential role in the oxaliplatin-induced elevation of 5-HT, ECL cells were visualised in the gastric mucosa using immunohistochemical detection of 5-HT. Similar to electrochemical recordings of 5-HT, more than double the number of ECL cells were observed in the gastric mucosa of oxaliplatin-treated mice (1.45 ± 0.17 ECL cells/gastric pit) compared to sham-treated mice (0.64 ± 0.03 ECL cells/gastric pit; *p* < 0.01) (n = 4 mice/group; Figure 7D).

## 4. Discussion

In this study, oxaliplatin-induced nausea and damage to the stomach were studied in mice and rats. In these models, oxaliplatin treatment: (i) induces pica as indicated by an increase in kaolin intake, decreases in food intake, and a decrease in body weight gain; (ii) increases gastric transit time represented by an increase in the stomach size, full stomach and stomach content weight, and a decrease in the empty stomach weight; (iii) causes gastric glands distortion, epithelial exfoliation, and muscle hypertrophy; (iv) significantly reduces the total number of myenteric and nNOS-IR neurons per surface area; (v) results in a significant reduction in the density of sympathetic, parasympathetic, and sensory nerve fibres in both the myenteric plexus and mucosa of the stomach; (vi) significantly increases parasympathetic nerve activity as indicated by an increase in VNf; and (vii) causes a significant increase in a 5-HT release from the increased number of ECL cells in the stomach.

Pica involves an alteration in feeding behaviour consisting of the ingestion of non-nutritive materials such as bark, clay (kaolin), and wood as a curative behavioural response for GI disturbances. The behavioural parameter, e.g., pica, is used as an indirect marker of nausea and vomiting and/or emesis in rodents to study the effects of emetogenic substances including chemotherapeutic agents [15]. Our study demonstrated that oxaliplatin treatment induces pica in mice. However, another study reported no alteration in the kaolin intake in mice after cisplatin injection [26]. A possible explanation for this discrepancy may include the observation period of only 2 days, whereas pica was found to be more profound over time and was highest at the final time point of two weeks, according to our observations. Several studies have reported on the effects of cisplatin on pica in rats. Cisplatin treatment induces pica and chemotherapy-induced nausea and vomiting [27]. Repeated administration of cisplatin in rats causes a significant reduction in food intake and body weight, and an increase in kaolin intake [14,28]. Increases in kaolin intake may occur within 24 h post-treatment [28]. Significant increases in kaolin intake were observed 3 days after the start of the oxaliplatin treatment in our study. Consistent with previous findings [14,28], our results also show that oxaliplatin treatment significantly decreases body weight gain and food intake.

Nausea and vomiting are the two most frequent and common GI side effects of chemotherapy and are multifactorial in their origin. It has been proposed that delayed gastric transit and gastric retention are contributing factors to nausea and vomiting [28]. Gastric transit/emptying can be delayed due to structural, neural, or humoral abnormalities [29]. Alteration in the sympathetic and parasympathetic nervous system is another important contributing factor to abnormalities in GI function, leading to nausea and vomiting. Cisplatin administration at both higher (6 mg/kg) and lower (3 mg/kg) doses significantly induced delayed gastric emptying and an increase in the stomach size and distension in rats [28]. Cisplatin treatment also increased the stomach size in mice [30]. In this study, similar results can be extrapolated with oxaliplatin-induced delays in gastric emptying. A significant increase in full stomach weight and stomach content weight is found without an increase in combined chow and kaolin consumption, which suggests that delayed gastric emptying is evident with the significant gastric content retention and distension. The distension can generate afferent impulses to, and stimulate, the vomiting centre [8]. Exclusively, satiation signals originate from the stomach in response to volumetric distension [31]. The phenomena of nausea and vomiting can be induced by ‘distension-induced anorexia’ that is mediated by both gastric and hepatic vagal branches [32], or ‘chemotherapy-induced pica or anorexia’ mediated by the hepatic vagal branch [33].

A histological assessment of the stomach mucosa established that significant mucosal damage with muscle hypertrophy and epithelial exfoliation occurs at the early stages of oxaliplatin administration. The damage becomes evident to the mucosal epithelium and luminal debris on day 3. It has been suggested that such a persistent mucosal insult may allow the passage of enterotoxins resulting in damage to the cells, exposing and inducing damage to enteric neurons and neuronal death [16,34]. A marked increase in fibrosis in both the muscle layers and myenteric plexus was observed in diabetic gastroparesis [29,35], along with reduced gastric emptying and neuronal loss. Structural damage to the mucosal barrier may make the stomach more favourable for other toxins to cause further damage as evidenced by the activation of 5-HT_3_ receptors by luminal toxins such as bile acids, ammonia, and ingested poisons that in turn induced emesis [36].

The total number of myenteric neurons as well as their nitrergic subpopulation was decreased in the stomach. This study is the first to investigate changes in the neuronal population in the stomach after oxaliplatin treatment. This finding is consistent with previous studies that reported a significant reduction in myenteric neurons and their subpopulations in the colon of mice by oxaliplatin [17,37], 5-fluorouracil [38], or cisplatin [30]. The neuronal NOS enzyme is responsible for the synthesis of nitric oxide (NO), the free-radical gas and inhibitory neurotransmitter, to the muscles of the GI tract and controls their relaxation, assisting in the passage of gastric contents [39,40]. Coordination between the excitatory and inhibitory neurotransmitters is necessary for the proper functioning of the stomach. Reduction in the number of myenteric and nNOS-IR neurons has been observed in patients with diabetic gastroparesis [29,35] and in an animal model of diabetes [41]. A loss of nitrergic neurons is suggested to be correlated with the pathogenesis of delayed gastric emptying [35]. In this study, we assessed the number of nNOS-IR neurons in the gastric corpus. The delayed gastric emptying and the subsequent increase in the stomach weight and content might be due to the changes in the proportion between the cholinergic and nitrergic neurons in the different stomach sub-regions. Determining the proportion between cholinergic and nitrergic neurons in the different stomach sub-regions in future studies is warranted.

The stomach receives dense inputs from sympathetic, parasympathetic, and sensory nerves that play a critical role in the regulation and modulation of GI function, including emesis and gastric contractile activity, via coordination between their excitatory and inhibitory functions [42]. Removal of the extrinsic innervation to the stomach leads to disorganised and dysregulated gastric activity that frequently results in symptoms such as nausea and vomiting. This study, for the first time, reveals a significant decrease in the density of TH-IR sympathetic, VAChT-IR parasympathetic cholinergic, and CGRP-IR sensory fibres in both the myenteric plexus and mucosa of the stomach following the oxaliplatin treatment in mice. The oxaliplatin-induced reduction in VAChT-IR fibres in the stomach might involve two possible mechanisms, including (i) oxaliplatin-induced inflammation of the stomach [43] as evidenced by the histological observations in our study, or (ii) an accumulation of platinum in the stomach causing direct cytotoxic effects. We have observed a significant amount of platinum accumulation within the enteric neurons in mice after chronic oxaliplatin treatment [44]. Previous studies suggested a correlation between ENS damage and the long-term changes in GI functions in diabetes [45] and GI inflammation [46,47,48]. However, only a few studies investigated alterations in the fibres innervating the stomach in diseased conditions with no studies on oxaliplatin-induced gastric changes. The reduction in TH-IR fibres was found in patients with diabetic gastroparesis [35] and in the aged population [49]. Further studies are required to reveal the mechanism of this altered extrinsic input and its correlated functional impact.

Sensory fibres originating from the dorsal root ganglia (DRG) in the spinal cord innervate the entire stomach [50] and regulate its function. The reduced density in the CGRP-IR sensory afferent nerve fibres in the myenteric plexus and mucosa of the stomach, observed in the study, is consistent with a previous report showing a decrease in CGRP-IR fibres in the rat intestine following cisplatin treatment [27]. The reduction in CGRP-IR fibres can be correlated with the oxaliplatin-induced damage of the DRG neurons [51,52,53] or to platinum-induced alteration in the axons and myelin sheathes of the neurons [54]. These findings propose that the GI sensation and the essential response to environmental factors are impeded due to a reduction in the density of sensory afferent fibres and the release of CGRP [55] from the myenteric plexus into the gastric mucosa. The oxaliplatin-induced loss of sensory afferent fibres, observed in the study, can be correlated with acute and chronic peripheral neuropathy of oxaliplatin chemotherapy.

The functional experiment in the study shows a significant increase in the vagus efferent nerve activity after the chronic oxaliplatin treatment in rats. This is the first study to investigate the effects of oxaliplatin on the efferent arm of the vagus nerve. The dorsal motor nucleus of the vagus mediates the efferent or motor function of the GI tract in the process of vomiting. During emesis, the cholinergic dorsal motor nucleus of the vagus activates nicotinic acetylcholine receptors within the stomach and upper GI tract [10]. Therefore, an oxaliplatin-induced increase in VNA may be correlated with its effect on the brainstem regulating parasympathetic function. Emesis is also accompanied by bradycardia, a reduction in the heart rate induced by increased parasympathetic activity. Augmentation of the cardiac parasympathetic tone was reported in subjects that have vomited [56,57]. Platinum-based drugs have also been implicated with cardiac toxicity, causing diastolic dysfunctions, hypertension, and myocardial ischemia [58,59,60], where the oxaliplatin-induced potentiation of vagal activity may play a role.

The gut serotonin (5-HT) plays a critical role in chemotherapy-induced nausea and vomiting. The 5-HT is released from the ECL cells and activates vagal afferent fibres via binding to the 5-HT_3_ receptors [61,62]. In this study, oxaliplatin treatment significantly increases the number of enterochromaffin-like cells as well as the release of 5-HT in the gastric mucosa. This finding is consistent with previous studies showing the release of 5-HT from ECL cells and activation of 5-HT_3_ receptors on vagal afferents by treatment with cisplatin and related drugs [63,64]. Cytotoxic drug-induced vomiting was reduced by vagotomy and 5-HT_3_ receptor antagonists [63,65], indicating the participation of vagal afferents as well as 5-HT in chemotherapy-induced nausea and vomiting. In addition, 5-fluorouracil treatment increased the expression of 5-HT_3_ receptors in nerve fibres in mice [66]. Nausea and vomiting were also induced by several infectious agents, including the rotavirus, cholera toxin, and *Campylobacter*, and the activation of vagal afferents via 5-HT released from ECL cells has been proposed to be involved in this process [67,68]. However, the mechanism of the oxaliplatin-induced 5-HT release from ECL cells is not known. Our data suggest that hyperplasia of ECL cells could play a role in the augmented 5-HT release from the gastric mucosa. Oxaliplatin-induced mitochondrial damage may be correlated with cell damage and the subsequent release of 5-HT [69]. Further study on the mechanism of oxaliplatin-induced 5-HT release is required.

In conclusion, we identified multiple structural, neurochemical, and functional changes in the stomach induced by oxaliplatin treatment that suggest multifactorial mechanisms underlying chemotherapy-induced GI disorders, including nausea and vomiting. While further studies are required to investigate the effects of oxaliplatin on enteric functions as well as the mechanisms of oxaliplatin-induced changes, our results indicate that the persistent delay in the gastric transit can be associated with enteric neuropathy affecting intrinsic neurons and extrinsic nerve fibres. Gastric distortion, increased kaolin consumption (pica), increased release of 5-HT, and potentiation of parasympathetic nerve activity may be correlated with chemotherapy-induced nausea and vomiting pathways. Further studies should be performed on morphological, immunohistochemical, and functional changes in the duodenum, taking into account the important role of the duodenum in the emetic reflex.

## Figures and Tables

**Figure 1 biomolecules-13-00276-f001:**
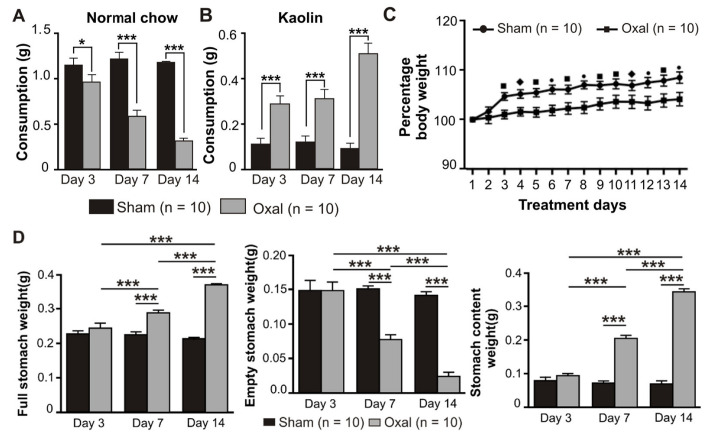
Effects of repeated oxaliplatin administration on pica, body weight, and stomach content. (**A**,**B**) Group data showing the consumption of normal chow (**A**) and kaolin (**B**) for 24 h over 14 days following oxaliplatin administration. A significant decrease in normal chow consumption while a significant increase in kaolin consumption was observed in the oxaliplatin-treated group. (**C**) The percentage of body weight decreased over time, starting from day 3 up to day 14 in chronic oxaliplatin-treated mice. (**D**) Group data showing the full stomach, empty stomach, and stomach content weight at days 3, 7, and 14 following oxaliplatin injection. Data are presented as the mean ± SEM. The number of animals is shown in parentheses. •, *** *p* < 0.001, ■ *p* < 0.001, ♦, * *p* < 0.05.

**Figure 2 biomolecules-13-00276-f002:**
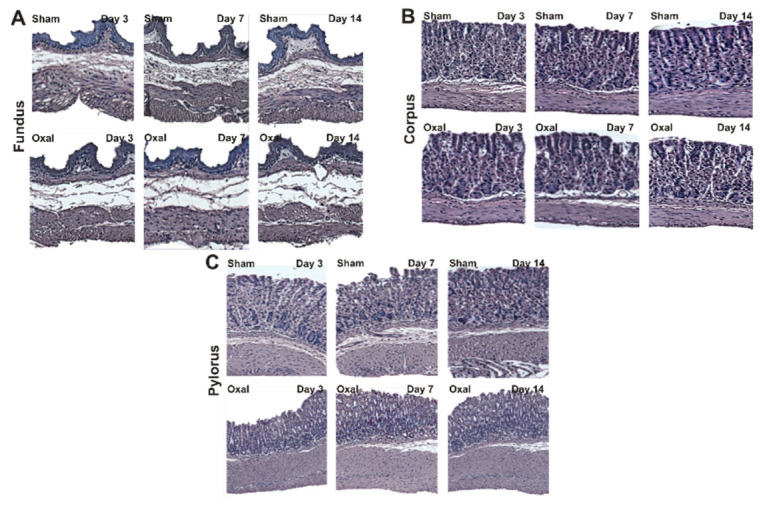
Histological assessment of changes in the morphology of the stomach in chronic oxaliplatin treated mice. (**A**–**C**) Hematoxylin and eosin staining of the fundus (**A**), corpus (**B**), and pylorus (**C**) at days 3, 7, and 14 following repeated oxaliplatin injection and sham treatment.

**Figure 3 biomolecules-13-00276-f003:**
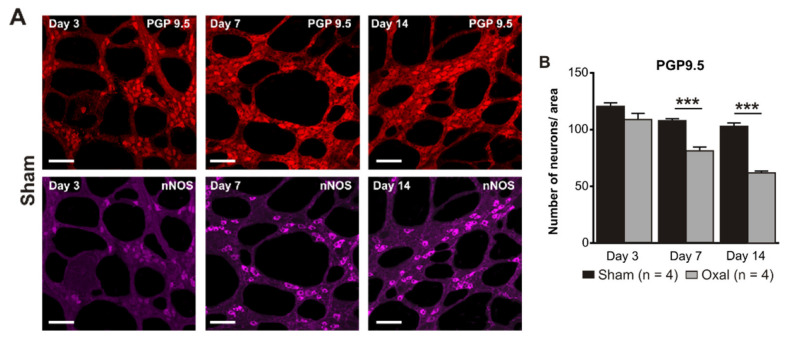
Effects of repeated oxaliplatin administration on the myenteric neurons and inhibitory motor neurons in whole mount preparations of the stomach. (**A**) Myenteric neurons labelled with the pan-neuronal marker PGP9.5 (red) and (**C**) inhibitory motor neurons labelled with the antibody for neuronal nitric oxide synthase (nNOS) (purple) in the gastric corpus at days 3, 7, and 14 following the injection of sterile water (sham) and oxaliplatin (oxal). Scale bar: 100 μm. (**B**,**D**) Group data showing the average number of PGP9.5-IR neurons and nNOS-IR neurons counted per a 0.25 mm^2^ area in wholemount preparations of the gastric corpus of sham and oxal-treated mice. Data are presented as the mean ± SEM. The number of animals is shown in parentheses. *** *p* < 0.001; * *p* < 0.05.

**Figure 4 biomolecules-13-00276-f004:**
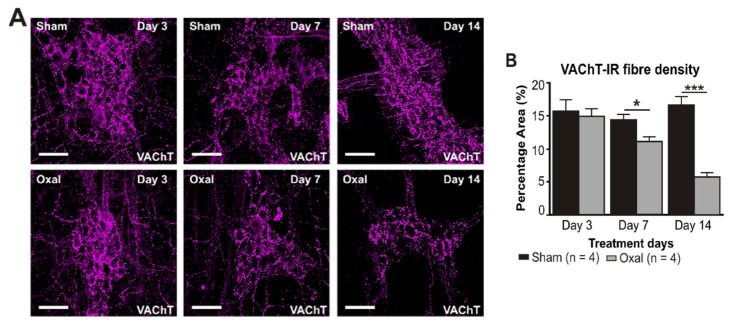
Effects of repeated oxaliplatin administration on cholinergic fibres in the stomach. (**A**) Cholinergic fibres labelled with the vesicular acetylcholine transporter (VAChT) in wholemount preparations of myenteric ganglia at days 3, 7, and 14 following the injection of sterile water (sham) and oxaliplatin (oxal). Scale bar: 100 μm. (**B**) Group data showing the density of VAChT-IR fibres per 0.25 mm^2^ area in wholemount preparations of the gastric corpus of sham- and oxal-treated mice. (**C**) Cholinergic fibres labelled with VAChT in cross-section preparations of the gastric corpus at day 14 following the injection of sterile water (sham) and oxaliplatin (oxal). Scale bar: 100 μm. (**D**) Group data showing the density of VAChT-IR fibres in cross-section preparations of the stomach of sham- and oxal-treated mice. Data are presented as the mean ± SEM. The number of animals is shown in parentheses. *** *p* < 0.001; * *p* < 0.05.

**Figure 5 biomolecules-13-00276-f005:**
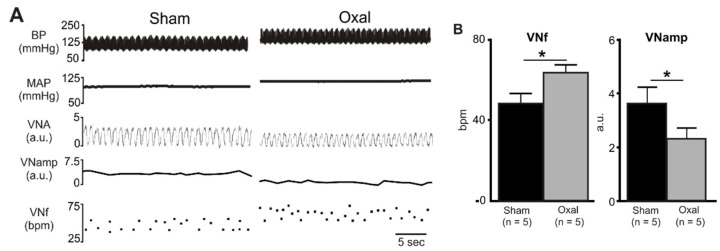
Effects of chronic oxaliplatin treatment on cholinergic nerve activity. (**A**) A representative recording of blood pressure (BP) (pulsatile and mean), vagus nerve activity (VNA) (rectified) [arbitrary units (a.u.)], vagus nerve amplitude (VNamp), and vagus nerve frequency (VNf) in vehicle-treated and chronic oxaliplatin-treated rats. (**B**) Grouped data of basal VNf and VNamp in vehicle-treated and oxaliplatin-treated rats. Effects are shown as absolute (VNf (bpm, bursts per minute)) or arbitrary (VNA, VNamp) values. Data are presented as the mean ± SEM. The number of animals is shown in parentheses. * *p* < 0.05 compared to the vehicle-treated group.

**Figure 6 biomolecules-13-00276-f006:**
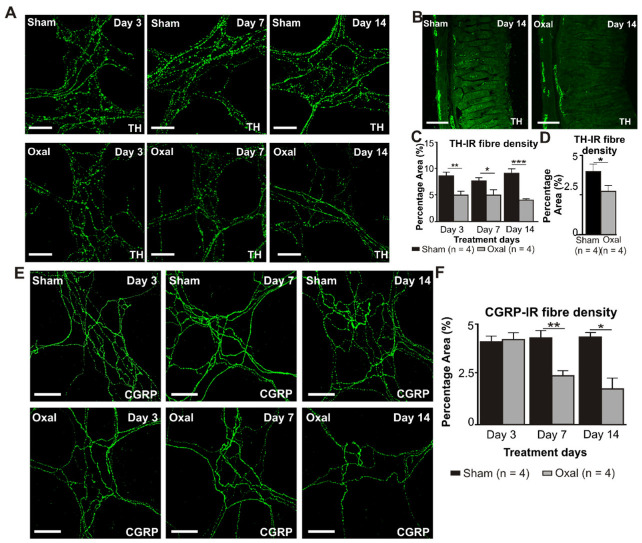
Effects of chronic oxaliplatin treatment on noradrenergic and sensory nerve fibres in the stomach. (**A**) Noradrenergic fibres labelled with tyrosine hydroxylase (TH) in wholemount preparations of myenteric ganglia of the gastric corpus at days 3, 7, and 14 following the injection of sterile water (sham) and oxaliplatin (oxal). Scale bar: 100 μm. (**B**) Noradrenergic fibres labelled with TH in cross-section preparations of the gastric corpus at day 14 following the injection of sterile water (sham) and oxaliplatin (oxal). Scale bar: 100 μm. (**C**) Group data showing the density of the TH-IR fibres per 0.25 mm^2^ area in wholemount preparations of the gastric corpus of sham and oxal-treated mice. (**D**) Group data showing the density of the TH-IR fibres in cross-section preparations of the gastric corpus of sham and oxal-treated mice. (**E**) Sensory nerve fibres labelled with calcitonin gene-related peptide (CGRP) in wholemount preparations of myenteric ganglia at days 3, 7, and 14 following the injection of sterile water (sham) and oxaliplatin (oxal). Scale bar: 100 μm. (**F**) Group data showing the density of the CGRP-IR fibres per 0.25 mm^2^ area in wholemount preparations of the stomach of sham and oxal-treated mice. Data are presented as the mean ± SEM. The number of animals is shown in parentheses. *** *p* < 0.001; ** *p* < 0.01; * *p* < 0.05.

**Figure 7 biomolecules-13-00276-f007:**
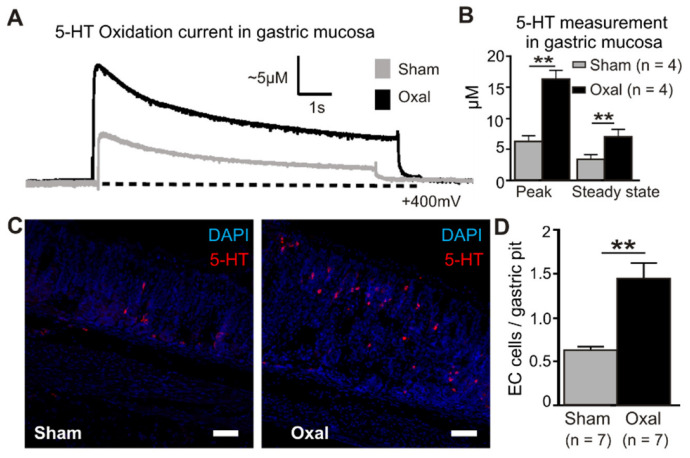
Effects of chronic oxaliplatin treatment on serotonin levels in the stomach. (**A**) Representative amperometric traces of the current generated by serotonin (5-hydroxy tryptamine (5-HT)) oxidation at the mucosal surface of the stomach. The current (nA) was converted to 5-HT (µM) by calibrating carbon fibre electrodes with a 10 µM 5-HT solution, as depicted in vertical scales. Mechanical stimulation of the gastric mucosa produced a compression-evoked release (peak) of 5-HT, which decayed back to basal levels (steady state) in both sham (grey traces) and oxal-treated mice (black traces). The dotted lines represent the baseline. (**B**) Group data comparing ‘peak’ and ‘steady state’ 5-HT levels between sham and oxal-treated mice. (**C**) ECL cells visualised by fluorescent immunohistochemical detection of 5-HT (red) in the mucosa of the gastric body (scale bar = 50 µm). (**D**) Quantification of ECL cells (5-HT + ve) in the gastric body of sham- and oxal-treated mice normalised per number of gastric pits. Data are presented as the mean ± SEM. The number of animals is shown in parentheses. ** *p* < 0.01.

## Data Availability

The data presented in this study are available on request from the corresponding author.

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
