# Peer review of "Oxaliplatin-Induced Damage to the Gastric Innervation: Role in Nausea and Vomiting"

_biomolecules, 2023, doi:10.3390/biom13020276_

Round 1

Reviewer 1 Report

Review on manuscript biomolecules-2052197: Oxaliplatin-induced damage to the gastric innervation: role in nausea and vomiting

Summary:

This study focusses in vivo effects of the chemotherapeutical drug Oxaliplatin, which is widely used for the treatment of colorectal cancer and which has been reported to have negative gastrointestinal side effects such as nausea and vomiting. The authors aimed to evaluate possible explanations for such complaints by investigating structural, neurochemical and functional changes of repeated intraperitoneal Oxaliplatin administration to mice and rats. They looked at the eating behaviour, including abnormal behaviour such as pica, stomach content, intrinsic gastric neuronal population, extrinsic innervation, levels of mucosal serotonin (5-HT) and parasympathetic vagal efferent nerve activity.

The main findings of the study are: (1) it induces pica and bodyweight loss, (2) it increased stomach weight and it’s content, hinting on a delayed gastric emptying, (3) it decreased the total number of myenteric and nitric oxide synthase-immunoreactive neurons as well as the density of gastric sympathetic, parasympathetic and sensory fibers, while it (4) increased levels of mucosal 5-HT and the number of enterochromaffin-like cells. With their results, the authors provide important additive knowledge on the underlying mechanisms of chemotherapy-induced gastrointestinal dysfunction and related clinical symptoms, which may also contribute to the development of targeted accompanied therapies for affected patients.

Main impression of the article and general comments:

My main impression of the study is positive. The scientific justification for the conduction of the study is given by the fact that colorectal cancer affects 10% of all cancer patients and that Oxaliplatin belongs to the group of the most frequently used chemotherapeutic drugs with well-known gastrointestinal side effects, negatively affecting overall therapy outcomes and patients’ quality of life. By combining functional and structural methods, the authors convincingly demonstrate direct effects of Oxaliplatin treatment on in vivo parameters (including eating behaviour, feed intake and weight gain), as well as ex vivo effects on the stomach morphology and on structural and functional properties of the extrinsic and intrinsic gastric innervation. It should be pointed out that this study is, to my knowledge, the first study providing results on the effect of Oxaliplatin on the gastric intrinsic neuronal population. The authors show a reduction in the total number of myenteric neurons and a reduction of the most important inhibitory neuronal subpopulation, which is a very interesting result. However, it would be even more interesting to look at these data for the different sub regions of the stomach. Since the neuronal circuits underlining gastric motility as well as the proportion between cholinergic and nitrergic neurons are different in the different stomach sub-regions, it will be interesting to know if this effect is, for instance, confined to the gastric antrum. This could be discussed and it could also be an explanation for the delayed gastric emptying and the subsequent increase in the stomach weight and content in the study (line 491 and following). Another question that came up to my mind was if the authors had a look at the duodenal mucosa. As the emetic reflex could also start in the first part of the small intestine, it will be of interest to know if similar modifications occur there.

Specific comments:

Material and methods

This section is well written and the experimental setup is properly described in general, but there are some points I would like to mention:

Line 100-109: Animals: please explain why you used mice and rats.

Line 143- 147: Stomach contents: would it not be appropriate to correct the contents weight for its water content?

Line 169: 0.2% picric acid seem to be a really high final concentration, maybe check it

Line 259 – 262: 5-HT-measurements: how did you ensure that you always applied the same pressure/force onto the mucosal cells? Otherwise, it should be discussed that differences in the applied pressure may have stimulated more or less 5-HT secretion.

Line 263 following: in this section you should already describe that you recorded vagal and phrenic nerve activity since in the results section (including figure 5) you describe vagus nerve activity and phrenic nerve frequency

Statistical analysis:

Line 287: why do you indicate SEM instead of SD? Since readers are generally interested in knowing the variability within in a sample, descriptive data should be more precisely summarized with SD.

Why did you decide to use paired t-test and one way ANOVA, respectively? Since you investigated two distinct animal groups (control vs. treated) your measurements are not matched.

Results

Line 300 -302: I think that here you copied in the instructions for authors.

Line 308/Fig. 1A: I think that it would be more precise to calculate feed consumption per bodyweight.

Line 309/Fig. 1C: it would be better to use different symbols to indicate statistical significance, since it is a bit confusing that you used the same symbols for the two animal groups

Line 353: I guess you mean a neuronal surface area of 0.25 mm²?

Line 358-359: you already described in the material and methods section that the anti -VAChT antibody identifies cholinergic fibers, so this is somehow redundant.

Line 378: do you mean VNf? As written above: it would be better and easier to understand, if you already introduce the phrenic nerve in the M&M section

Line 381/Fig. 5: what do you exactly mean by “arbitrary units”?; you use “bpm” twice, as an abbreviation for the unit “beats per minute” and “bursts per minute”; either you should use a different one for the latter (what about burst frequency?) or you have to explain it at both sites.

Discussion

For this section I would just like to refer to my already mentioned points regarding the potential meaning of the reduced number of NOS positive neurons for gastric emptying and some additional discussion on the method to stimulate mucosal 5-HT secretion.

Figures

I suggest to increase the size of the immunohistochemistry picture in order to better compare them.

In conclusion, I recommend publication of the manuscript after implementing the suggested changes and addition of a data availability statement.

Author Response

Reviewer 1

Review on manuscript biomolecules-2052197: Oxaliplatin-induced damage to the gastric innervation: role in nausea and vomiting

Summary:

This study focusses in vivo effects of the chemotherapeutical drug Oxaliplatin, which is widely used for the treatment of colorectal cancer and which has been reported to have negative gastrointestinal side effects such as nausea and vomiting. The authors aimed to evaluate possible explanations for such complaints by investigating structural, neurochemical and functional changes of repeated intraperitoneal Oxaliplatin administration to mice and rats. They looked at the eating behaviour, including abnormal behaviour such as pica, stomach content, intrinsic gastric neuronal population, extrinsic innervation, levels of mucosal serotonin (5-HT) and parasympathetic vagal efferent nerve activity.

The main findings of the study are: (1) it induces pica and bodyweight loss, (2) it increased stomach weight and it’s content, hinting on a delayed gastric emptying, (3) it decreased the total number of myenteric and nitric oxide synthase-immunoreactive neurons as well as the density of gastric sympathetic, parasympathetic and sensory fibers, while it (4) increased levels of mucosal 5-HT and the number of enterochromaffin-like cells. With their results, the authors provide important additive knowledge on the underlying mechanisms of chemotherapy-induced gastrointestinal dysfunction and related clinical symptoms, which may also contribute to the development of targeted accompanied therapies for affected patients.

Main impression of the article and general comments:

My main impression of the study is positive. The scientific justification for the conduction of the study is given by the fact that colorectal cancer affects 10% of all cancer patients and that Oxaliplatin belongs to the group of the most frequently used chemotherapeutic drugs with well-known gastrointestinal side effects, negatively affecting overall therapy outcomes and patients’ quality of life. By combining functional and structural methods, the authors convincingly demonstrate direct effects of Oxaliplatin treatment on in vivo parameters (including eating behaviour, feed intake and weight gain), as well as ex vivo effects on the stomach morphology and on structural and functional properties of the extrinsic and intrinsic gastric innervation. It should be pointed out that this study is, to my knowledge, the first study providing results on the effect of Oxaliplatin on the gastric intrinsic neuronal population. The authors show a reduction in the total number of myenteric neurons and a reduction of the most important inhibitory neuronal subpopulation, which is a very interesting result. However, it would be even more interesting to look at these data for the different sub regions of the stomach. Since the neuronal circuits underlining gastric motility as well as the proportion between cholinergic and nitrergic neurons are different in the different stomach sub-regions, it will be interesting to know if this effect is, for instance, confined to the gastric antrum. This could be discussed and it could also be an explanation for the delayed gastric emptying and the subsequent increase in the stomach weight and content in the study (line 491 and following). Another question that came up to my mind was if the authors had a look at the duodenal mucosa. As the emetic reflex could also start in the first part of the small intestine, it will be of interest to know if similar modifications occur there.

Response: We are thankful to the reviewer for these comments. We have done a histological assessment of the pylorus, corpus and fundus but immunohistochemistry was done only for the corpus. We agree with the Reviewer that the delayed gastric emptying and the subsequent increase in the stomach weight and content might be due to the changes in the proportion between cholinergic and nitrergic neurons in the different stomach sub-regions. Determining the proportion between cholinergic and nitrergic neurons in the different stomach sub-regions will be our next target. We have added this to the Discussion (lines 514-519).  We also agree with the Reviewer regarding the role of the duodenum in the emetic reflex. We are planning to perform morphological, immunohistochemical and functional studies of the duodenum. This has been added to the Discussion (lines 591-594).

Specific comments:

Material and methods

This section is well written and the experimental setup is properly described in general, but there are some points I would like to mention:

Line 100-109: Animals: please explain why you used mice and rats.

Response: We used both mice and rats due to the techniques used in this study. The electrophysiological experiments carried out in this study (isolation and recording of vagus nerve activity in particular) are very difficult to perform in mice due to the small animal size. Therefore, electrophysiological experiments were done in rats.

Line 143- 147: Stomach contents: would it not be appropriate to correct the contents weight for its water content?

Response: We measured the dry weight of the stomach’s solid content. We clarified this in the text (line 147).

Line 169: 0.2% picric acid seem to be a really high final concentration, maybe check it

Response: Zamboni’s fixative containing 0.2% picric acid and 2% paraformaldehyde in 0.1-M sodium phosphate buffer is commonly used for immunohistochemistry by other labs (e.g. Di Natale MR, Hunne B, Liew JJM, Fothergill LJ, Stebbing MJ, Furness JB. Morphologies, dimensions and targets of gastric nitric oxide synthase neurons. Cell Tissue Res. 2022;388(1):19-32. PMID: 35146560).  This concentration of picric acid inhibits autofluorescence of enteric neurons. We published several papers using this fixative (Rahman et al, Cell Tissue Res. 2016;366(2):285-299, McQuade et al, Br J Pharmacol, 2016. 173(24): p. 3502-3521, Robinson et al, J Histochem & Cytochem, 2016. 64(9): p. 530-545).  We have added references to the Methods section (line 171).

Line 259 – 262: 5-HT-measurements: how did you ensure that you always applied the same pressure/force onto the mucosal cells? Otherwise, it should be discussed that differences in the applied pressure may have stimulated more or less 5-HT secretion.

Response: The carbon fibre microelectrode was mounted on a precision micromanipulator (World Precision Instruments, Sarasota, FL, USA). The micromanipulator provides fine mechanical placement of the microelectrode with movement as little as a micron at a time. Therefore, the force applied to compress the mucosa was consistent.  Moreover, the same investigator conducted all experiments in a blinded manner. We have added this info to the Methods section (lines 263-266).

Line 263 following: in this section you should already describe that you recorded vagal and phrenic nerve activity since in the results section (including figure 5) you describe vagus nerve activity and phrenic nerve frequency

Response: Thank you very much for noticing this discrepancy. We only recorded vagus nerve activity. Vagus nerve frequency and amplitude were derived from vagus nerve activity. We have corrected the Figure 5 legend.

Statistical analysis:

Line 287: why do you indicate SEM instead of SD? Since readers are generally interested in knowing the variability within in a sample, descriptive data should be more precisely summarized with SD.

Response: We choose to use SEM over SD because it shows both SD and the accuracy of mean computation.

Why did you decide to use paired t-test and one way ANOVA, respectively? Since you investigated two distinct animal groups (control vs. treated) your measurements are not matched.

Response: Thanks for noticing this discrepancy. The paired t-test was an error. We used the unpaired t-test to analyze the difference in the number of ECL cells, the number of neurons per area, the percentage of the area of fiber expression between all groups, and the parameters of electrophysiology experiments. One-way analysis of variance followed by Bonferroni’s multiple group comparison test was used to analyze the weight of the stomach and its content at different time points. We have corrected this in the manuscript (lines 292-298).

Results

Line 300 -302: I think that here you copied in the instructions for authors.

Response: Thanks for noticing this. Apologies for that. It was overlooked while checking. We have deleted this in the manuscript (lines 308-310).

Line 308/Fig. 1A: I think that it would be more precise to per bodyweight.

Response: We used male Balb/c mice of the same body weight and age for the food/kaolin consumption experiments (n=10/group). The deviation of the body weight within each group at various time points was not significant, therefore, it would not affect the results. We have added these details to the text (lines 313-14 and 327-28).

Line 309/Fig. 1C: it would be better to use different symbols to indicate statistical significance, since it is a bit confusing that you used the same symbols for the two animal groups

Response: We used different symbols for different P values while comparing body weight between sham-treated (a line with circles) and oxaliplatin-treated (a line with squares) groups from day 1 to day 14 in Figure 1C. The description for different symbols is provided in the figure legend.

Line 353: I guess you mean a neuronal surface area of 0.25 mm²?

Response: We analysed the number of neurons within a 0.25 mm2 area. This has been corrected in the text (line 365).

Line 358-359: you already described in the material and methods section that the anti -VAChT antibody identifies cholinergic fibers, so this is somehow redundant.

Response:  We agree with the Reviewer; we deleted this sentence (lines 370-371). Thank you.

Line 378: do you mean

Response: As we mentioned above, we only recorded vagus nerve activity, not phrenic nerve. Vagus nerve amplitude (VNamp) and frequency (VNf) were derived from the vagus nerve activity (VNA).

Line 381/Fig. 5: what do you exactly mean by “arbitrary units”?; you use “bpm” twice, as an abbreviation for the unit “beats per minute” and “bursts per minute”; either you should use a different one for the latter (what about burst frequency?) or you have to explain it at both sites.

Response: We use the arbitrary unit for nerve activity as because during/before recording we calibrate nerve activity between +100 and -100. To avoid confusion with “beats per minute” and “bursts per minute” abbreviated as “bpm” units, we have deleted the heart rate (HR) recording from Figure 5.

Discussion

For this section I would just like to refer to my already mentioned points regarding the potential meaning of the reduced number of NOS positive neurons for gastric emptying and some additional discussion on the method to stimulate mucosal 5-HT secretion.

Response:  We have added this to the Discussion (lines 514-519) and to the Methods (lines 263-266).

Figures

I suggest to increase the size of the immunohistochemistry picture in order to better compare them.

Response:  We have increased the size of all immunohistochemical and histological images.

In conclusion, I recommend publication of the manuscript after implementing the suggested changes and addition of a data availability statement.

Response: We are thankful to the Reviewer for their constructive comments. We have addressed all comments and made changes to the manuscript and figures. We hope that these changes improved the quality of this submission. We have provided the data availability statement.

Reviewer 2 Report

The presented study “Oxaliplatin-induced damage to the gastric innervation: role in nusea and vomiting” was properly designed, and well prepared, and it is a very interesting experimental study concerned with gastric dysfunction during and after chemotherapy treatment due to colon cancer. This study is worth to be published in the “Biomolecules” journal.  

1.Oxaliplatin treatment is widely used during chemotherapy.

Patients suffer from the serious side effects of this medicine.  Between others, nausea and vomiting are substantial and commonly present. The experimental studies (animal experiments) should explain the basic mechanism concerned with this phenomenon.

2.This study gives fundamental information which may be further developed in the next investigations. It is relevant at the level of basic knowledge.

3.This study was properly designed, and many laboratory methods were used to elicit reasons of the side effects of implemented/potential chemotherapy treatment.

4. The used methodology very evidently demonstrates the oxaliplatin impact on stomach dysfunction, and it is done at this experimental level.

 5. The conclusions consistent with the evidence and arguments presented and do they address the main question posed.

6.  The references are contemporary and very well correspond with the results obtained.

7. Prepared illustrations very clearly demonstrate obtain results, they are very carefully prepared.

Author Response

Response: We are thankful to the Reviewer for their positive comments.